# WEAKLY SEMI-SUPERVISED NEURAL TOPIC MODELS

**Ian Gemp**[*]
DeepMind, London, UK
imgemp@google.com

**Ramesh Nallapati, Ran Ding, Feng Nan, Bing Xiang**
Amazon Web Services, New York, NY 10001, USA
{rnallapa,rding,nanfen,bxiang}@amazon.com

## ABSTRACT

We consider the problem of topic modeling in a *weakly semi-supervised* setting. In this scenario, we assume that the user knows a priori a subset of the topics she wants the model to learn and is able to provide a few exemplar documents for those topics. In addition, while each document may typically consist of multiple topics, we do not assume that the user will identify all its topics exhaustively.

Recent state-of-the-art topic models such as NVDM, referred to herein as Neural Topic Models (NTMs), fall under the variational autoencoder framework. We extend NTMs to the weakly semi-supervised setting by using informative priors in the training objective. After analyzing the effect of informative priors, we propose a simple modification of the NVDM model using a logit-normal posterior that we show achieves better alignment to user-desired topics versus other NTM models.

## 1 INTRODUCTION

Topic models are probabilistic models of data that assume an abstract set of topics underlies the data generating process (Blei, 2012). These abstract topics are often not only useful as feature representations for downstream tasks, but also for exploring and analyzing a corpus. Topic models are used to explore natural scenes in images (Fei-Fei and Perona, 2005; Mimno, 2012), genetics data (Rogers et al., 2005), and numerous text corpora (Newman et al., 2006; Mimno and McCallum, 2007).

While latent Dirichlet allocation (LDA) serves as the classical benchmark for topic models, recent state-of-the-art topic models such as NVDM (Miao et al., 2016) fall under the variational autoencoder (VAE) framework (Kingma and Welling, 2013), which we refer to as Neural Topic Models (NTMs). NTMs leverage the flexibility of deep learning to fit an approximate posterior using variational inference. This posterior can then be used to efficiently predict the topics contained in a document. NTMs have been shown to model documents well, as well as associate a set of meaningful top words with each topic (Miao et al., 2017).

Often, the top words associated with each extracted topic only approximately match the user's intuition. Therefore, the user may want to guide the model towards learning topics that better align with natural semantics by providing example documents. To our knowledge, supervision has been explored in more classical LDA models, but has not been explored yet in the NTM literature.

Labeling the existence of topics in each document across a corpus is prohibitively expensive. Hence, we focus on a weak form of supervision. Specifically, we assume a user may identify the existence of a single topic in a document. Furthermore, if a user does not specify the existence of a topic, it does not mean the topic does not appear in the document.

The main contribution of our work is an NTM with the ability to leverage minimal user supervision to better align topics to desired semantics.

## 2 BACKGROUND: NEURAL TOPIC MODELS

Topic models describe documents (typically represented as bags-of-words) as being generated by a mixture of underlying abstract topics. Each topic is represented as a distribution over the words in

---

[*]Work done while interning at Amazon NYC in Summer 2018.

a vocabulary so that each document can be expressed as a mixture of different topics and its words drawn from the associated mixture distribution.

A Neural Topic Model is a topic model constructed according to the variational autoencoder (VAE) paradigm. The generative process for a document $x$ given a latent topic mixture $v$ is $v \sim p(v), x \sim p_\theta(x|v)$, where $p(v)$ is the prior and $\theta$ are learned parameters. The marginal likelihood $p_\theta(x)$ and posterior $p_\theta(v|x)$ are in general intractable so a standard approach is to introduce an approximate posterior, $q_\phi(v|x)$, and seek maximization of the evidence lower bound (ELBO):

$$\log p_\theta(x) \geq \underbrace{\mathbb{E}_{q_\phi(v|x)}\big[\log p_\theta(x|v)\big] - D_{KL}(q_\phi(v|x)||p(v))}_{\text{ELBO}(x;\theta,\phi)}. \tag{1}$$

The reparamterization trick (Kingma and Welling, 2013) lets us train this model via stochastic optimization. The most commonly used reparameterization is for sampling from a Gaussian with diagonal covariance, e.g., $v = \mu(x) + \sigma(x)\epsilon, \ \epsilon \sim \mathcal{N}(0,1)$ where $\mu(x)$ and $\sigma(x)$ are neural networks.

## 3 Weakly Semi-Supervised Topic Modeling

As motivated in the introduction, the topics extracted from unsupervised topic modeling can be challenging to interpret for a user (see Table 1). It is often the case that a user has a set of "ground truth" topics in mind that they would like the model to align with. We focus on the setting where a user is presented a subset of documents and identifies at least a single topic for each of the documents.

| (U) rf harvesting energy chains documents consumption compression shortest texts retrieval | (S) recommendation retrieval item document items filtering documents collaborative preferences engine |
| --- | --- |

Table 1: Upon scanning topics extracted by an unsupervised (U) NVDM for the AAPD ArXiv dataset, the user recognizes a few words relevant to the topic "information retrieval". Supervision (S) leveraging exemplar documents containing the topic helps the model to better align the topic to the desired semantics.

We emphasize two challenging components in our setting. First, it is *semi-supervised* in the sense that only a small subset of documents is labeled by the user. Second, the supervision is *weak* in the sense that the user does not necessarily identify all the topics that are present in a document.

We define "ground truth" word rankings for each topic using a relative chi-square association metric, a common measure of word association. Specifically, we compute the chi-square statistics between the occurrence of each word and occurrence of each document class using the ground truth topic labels. We then divide the chi-square statistic for each word by the average of its chi-square statistics over all labeled topics, giving us a relative importance that is discriminative across topics. The top-10 words for each topic are then the words with highest relative importance. To evaluate alignment of topics extracted by an NTM to these "ground truth" topics, we compute the average normalized point-wise mutual information (NPMI) (Bouma, 2009) between each word in the NTM's topics and each word in the "ground truth" topics—a higher score indicating the NTM's extracted words often co-occur with the ground truth's across Wikipedia. We refer to this score as pairwise-NPMI (P-NPMI). We additionally report the average NPMI computed between all word pairs within each NTM topic (NPMI) as a measure of topic coherence as well as NPMI separately averaged over just supervised topics (S-NPMI) and unsupervised topics (U-NPMI).

## 4 Our Approach

Our approach focuses on extending the NTM to the semi-supervised setting. Our method is based on modifying the prior, $p_i(v_i)$, of a VAE to incorporate user label information, where $p_i(v_i)$ is the prior distribution over the latent variables for the $i^{\text{th}}$ document. As discussed above, we assume the user provides partial binary labels indicating the existence of topics in a document (1=exists, 0=unlabeled). We would like to encourage the posterior samples of the model, $v_i \sim q_\phi(v_i|x_i)$, to match the true presence of topics in a document.

In the case of a Gaussian posterior (NVDM), a natural interpretation of $v_i$ is as the logits of the probabilities that each topic exists. Therefore, to recover probabilities, we simply `sigmoid` the outputs

of our encoder. Given this interpretation for semi-supervision, we set $p_i(v_{ik})$ to be $\mathcal{N}(\mu_{ik}, 1)$, where

$$
\mu_{ik} = \begin{cases} \texttt{logit}(0.9999) & \text{if document } i \text{ is labeled to} \\ & \text{contain topic } k, \\ \texttt{logit}(p_{k+}) & \text{else if topic } k \text{ is supervised,} \\ \texttt{logit}(0.5) = 0 & \text{otherwise,} \end{cases} \tag{2}
$$

and $p_{k+}$ is the class-prior probability, i.e. the probability that any given document contains topic $k$. Note that $p_{k+}$ can be estimated from PU data (Jain et al., 2016). In experiments, we instead use the mean class-prior probability for all topics ($\bar{p}_+$) and are still able to obtain high performance suggesting this approach is robust to estimation error.

In order to extract topics from the model, we condition the generative distribution, $p_\theta(x|v)$, on the logits of each of the possible approximate $\texttt{onehot}$ vectors, e.g., $v_1 = \texttt{logit}([0.9999, 0.0001, \ldots])$ for topic 1. After conditioning on $v_k$, we observe the mean of the conditional distribution, $\xi_k$. We then subtract from each $\xi_k$ the mean of $\xi$ over the topics, i.e., $\tilde{\xi}_k = \xi_k - \bar{\xi}$. The indices corresponding to the top-10 values of $\tilde{\xi}_k$ indicate the top-10 words associated with each topic $k$.

We also explore using two other posteriors. One is the Concrete / Gumbel-softmax approximation to the Bernoulli (Jang et al., 2016; Maddison et al., 2016) which more closely matches our goal of modeling the existence of topics in documents. The differentiable posterior $q_\phi(v|x) = Concrete(\pi, \tau)$ can be sampled with Equations (3), (4), and (5):

$$
\begin{aligned}
u &\sim U(0,1) & (3) \\
\eta &= \texttt{logit}(\pi) + \texttt{logit}(u) & (4) \\
v &= \texttt{sigmoid}(\eta/\tau) & (5)
\end{aligned}
\qquad
\pi_{ik} = \begin{cases} 0.9999 & \text{if document } i \text{ is labeled to} \\ & \text{contain topic } k, \\ p_{k+} & \text{else if topic } k \text{ is supervised,} \\ 0.5 & \text{otherwise.} \end{cases} \tag{6}
$$

where $\tau$ is a hyperparameter such that $v \sim Ber(\pi)$ as $\tau \to 0$, $\texttt{sigmoid}(r) = (1 + e^{-r})^{-1}$, and $\texttt{logit}(p) = \texttt{sigmoid}^{-1}(p) = \log(p/(1-p))$. The prior is $p(v_{ik}) \sim Concrete(\pi_{ik}, \tau)$ where $\pi_{ik}$ is the prior probability defined in Equation (6). The log-density of $Concrete(\pi, \tau)$ is given in Section C.3.2 of (Jang et al., 2016) and we use the Bernoulli-$D_{KL}$ as an approximation. The other posterior we consider is a logit-normal distribution, for which the $\texttt{logit}(v)$ is normally distributed. Samples are easily obtained by taking the sigmoid of samples from a normal distribution.

### COMPARISON OF KULLBACK LIEBLER DIVERGENCES

Our technique of incorporating supervision via informative priors provides a supervised training signal via the $D_{KL}$ term in Equation 1. Figure 1 compares the Bernoulli and Gaussian (NVDM) $D_{KL}$ terms against a standard cross-entropy loss. Note $D_{KL}$ is the same for Gaussian and logit-normal.

In Figure 1, we fix the prior distributions to $\texttt{Ber}(\pi_p = 0.9999)$ for the Bernoulli model and $\mathcal{N}(\mu_p = \texttt{logit}(0.9999), \sigma_p = 1)$ for NVDM to signify the presence of a topic with high certainty. The x-axis, $p$, denotes the posterior probability: $\pi_q$ for the Bernoulli model and $\texttt{sigmoid}(\mu_q)$ for NVDM/logit-normal. The cross entropy loss is $-\pi_p \log p - (1 - \pi_p) \log(1 - p)$ where $\pi_p$ is the prior probability. The posterior variance of the NVDM is assumed to match the prior variance in this figure. It is clear that NVDM $D_{KL}$ strictly provides the strongest loss signal for topic prediction.

On the other hand, the Bernoulli model generally achieves better topic coherence (NPMI). We expect this is due to the cleaner decoding process relative to NVDM. For example, the Bernoulli decoding computes the word distribution for the first out of $K$ topics as the softmax of $1 \cdot W_1 + \ldots + 0 \cdot W_K + b$ where $W \in \mathbb{R}^{|V| \times K}$ and $b \in \mathbb{R}^{|V|}$ are the decoder weight matrix and bias respectively, $|V|$ is the size of the vocabulary, and $W_k$ refers to the $k^{\text{th}}$ row of $W$. In contrast, NVDM decodes with $9.2 \cdot W_1 + \ldots - 9.2 \cdot W_K + b$ where $9.2 \approx \texttt{logit}(0.9999) = -\texttt{logit}(0.0001)$.

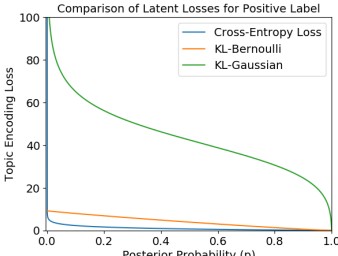

Figure 1: The Gaussian-$D_{KL}$ provides the strongest topic encoding signal. The Bernoulli $D_{KL}$ is approximately linear with finite divergence at $p = 0$. The log loss is very shallow except for where it diverges at $p = 0$.

Therefore, we propose NVDM-$\sigma$ with a logit-normal posterior. The intuition is to retain both the strong supervision signal of NVDM as well as the clean decoding used by the Bernoulli model.

## 5 Experiments

We train and evaluate these models on three multi-label datasets: *BibTeX*, a tagging dataset containing BibTeX metadata, *Delicious*, a document classification dataset, and *AAPD*, a dataset extracted from the abstracts and subjects of computer science ArXiv papers. We vary three experimental variables. We consider the percentage of topics supervised to be either 10%, 50%, or 100%. For example, for Delicious with 20 known topics, we consider supervising 2, 10, and all 20 dimensions of $v$. We also consider the number of provided labeled samples per supervised topic. For example, in one experiment, when supervising 10% of topics for Delicious, we consider providing 3 or 10 example documents per topic. Lastly, we vary the number of labels given per document. As described in Section 3, the user is not required to provide all the topics contained in each document. We consider documents with only 1 label, up to half the maximum number of labels (5 for Delicious), and fully-labeled (up to 11). Using 10% topic supervision, 10 labeled documents per topic, and 1 label per document as an experimental baseline, we vary each experimental variable independently resulting in six different settings for each of three datasets. Note each setting is still weakly semi-supervised.

In general, all models see a rise in NPMI from informative priors. We observe that NPMI rises on average by 12% relative to the unsupervised setting and by 15% beyond that to the fully supervised case. This demonstrates that weak labels are capable of providing a significant boost in performance.

| Model | Top-10 Words (First Iteration) | Top-10 Words (Fully Trained) |
|-------|-------------------------------|------------------------------|
| NVDM (0.230) | transformation software attributes np necessary correlation author posterior oracle alternating | word allocation string greedy distance jointly tree sub trees latent |
| Bernoulli (0.219) | provides received game white numbers distinct increasingly effectiveness loss gives | greedy performance procedure allocation distance jointly speed report existing learn |
| NVDM-$\sigma$ (0.239) | rf outperforms produces availability minimal equation correcting increased formalism centrality | allocation distance string greedy jointly correction word sub tree learn |

Table 2: The supervised topic for AAPD is *Computation and Language* and the P-NPMI scores are under the model names. Only 3 documents are provided for 10% of topics with 1 label per document.

Table 2 shows successful alignment for supervised topics. Only three documents are labeled for each topic; this positive result is important as labeling numerous documents is prohibitively expensive for commercial applications. Table 3 shows NVDM-$\sigma$ does best in terms of all NPMI metrics.

| Metric | NVDM | NVDM-$\sigma$ | Bernoulli |
|--------|------|---------------|-----------|
| P-NPMI | 0.852 (0.099) | **0.978** (0.048) | 0.892 (0.126) |
| NPMI | 0.888 (0.100) | **0.960** (0.052) | 0.947 (0.049) |
| U-NPMI | 0.915 (0.102) | **0.951** (0.055) | 0.935 (0.060) |
| S-NPMI | 0.878 (0.094) | **0.953** (0.045) | 0.951 (0.057) |

Table 3: Average performance (with standard deviation in parentheses) relative to best score in each experimental setting. Specifically, for each experimental setting and each metric, we choose the best score among the 3 models and normalize the scores according to it. We then average these relative scores over the experimental settings. Note NVMD-$\sigma$ achieves high mean performance with relatively low variance.

## 6 Conclusion

In this work, we proposed supervising Neural Topic Models with weak supervision via informative priors and explored a variety of model posteriors. A careful analysis of their KL divergences and decoding mechanisms led us to an NTM with logit-normal posterior which best aligned extracted topics to desired user semantics.

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

## A  MODEL ARCHITECTURE

We use a feedforward neural network with 2 hidden layers followed by a linear transformation for the encoder. The first layer is $3K$ neurons wide and the second is $2K$ neurons wide where $K$ is the size given to the latent space. Each hidden layer uses `sigmoid` nonlinearities. The final layer is linear and outputs a vector of size $2K$ for the parameters of the Gaussian posterior. The mean is obtained from the first $K$ entries of this vector and the standard deviation, $\sigma$, is obtained from the last $K$ entries by applying a softplus, $\log(1 + e^x)$. Instead of directly modeling the probability $\pi$ in Equation (5) for the Bernoulli posterior, we interpret the output of the final linear layer of the encoder as $\text{logit}(\pi)$. We set the dimensionality of the latent-space $\upsilon$ equal to 50 for 20NG and the number of ground truth topics for all other datasets. The decoder always consists of a linear layer followed by a `softmax` to obtain the multinomial probabilities for each word in the vocabulary. We set $\tau = 1$ for the Gumbel-softmax sampler. We train with Adadelta, rescale gradients by 0.01 (smaller), and use batch sizes of 256.

## B  SEMI-SUPERVISED TOPIC MODEL TEST PERFORMANCE

| Metric | Model | Bibtex | Delicious | ArXiv |
|---|---|---|---|---|
| PPX | NVDM | 15835 | 1955 | 1456 |
| | Bernoulli | **858** | **1683** | **1126** |
| | NVDM+sig | 19217 | 2175 | 1035 |
| $-\log(p(x|\upsilon))$ | NVDM | 450 | **894** | **428** |
| | Bernoulli | **441** | 900 | 446 |
| | NVDM+sig | 442 | 925 | 402 |
| P-NPMI | NVDM | 0.194 | **0.074** | 0.180 |
| | Bernoulli | **0.234** | 0.061 | **0.231** |
| | NVDM+sig | 0.204 | 0.061 | 0.223 |
| NPMI (Avg/S) | NVDM | 0.230/0.260 | 0.149/**0.236** | **0.214**/0.183 |
| | Bernoulli | 0.233/**0.267** | 0.153/0.217 | 0.211/**0.234** |
| | NVDM+sig | **0.238**/0.259 | **0.180**/0.211 | 0.196/0.205 |

Table 4: 10 Documents provided for 10% of Topics with 1 label per document.

| Metric | Model | Bibtex | Delicious | ArXiv |
|---|---|---|---|---|
| PPX | NVDM | $19 \times 10^9$ | 3149 | 9119 |
| | Bernoulli | **7864** | **1943** | **992** |
| | NVDM+sig | $7.08 \times 10^9$ | 4204 | 12332 |
| $-\log(p(x|\upsilon))$ | NVDM | 448 | **905** | 422 |
| | Bernoulli | 447 | 919 | **420** |
| | NVDM+sig | **443** | 918 | 431 |
| P-NPMI | NVDM | 0.214 | 0.058 | 0.211 |
| | Bernoulli | 0.197 | 0.060 | 0.228 |
| | NVDM+sig | **0.219** | **0.069** | **0.239** |
| NPMI (Avg/S) | NVDM | 0.240/0.219 | 0.124/0.114 | 0.217/0.190 |
| | Bernoulli | 0.222/0.225 | **0.140**/0.134 | 0.215/0.216 |
| | NVDM+sig | **0.251/0.249** | 0.137/**0.138** | **0.234/0.224** |

Table 5: 10 Documents provided for 50% of Topics with 1 label per document.

| Metric | Model | Bibtex | Delicious | ArXiv |
|---|---|---|---|---|
| PPX | NVDM | $92.5 \times 10^{12}$ | 17877 | 290359 |
| | Bernoulli | **24479** | **2042** | **1358** |
| | NVDM+sig | $18.6 \times 10^{13}$ | 12876 | 256033 |
| $-\log(p(x|v))$ | NVDM | 458 | 940 | **409** |
| | Bernoulli | **447** | 915 | 422 |
| | NVDM+sig | **447** | **904** | 415 |
| P-NPMI | NVDM | 0.206 | 0.048 | 0.215 |
| | Bernoulli | 0.201 | 0.051 | **0.235** |
| | NVDM+sig | **0.230** | **0.067** | **0.235** |
| NPMI (Avg/S) | NVDM | 0.216/0.216 | 0.104/0.104 | 0.184/0.184 |
| | Bernoulli | 0.226/0.226 | 0.139/0.139 | **0.244/0.244** |
| | NVDM+sig | **0.252/0.252** | **0.145/0.145** | 0.217/0.217 |

Table 6: 10 Documents provided for 100% of Topics with 1 label per document.

| Metric | Model | Bibtex | Delicious | ArXiv |
|---|---|---|---|---|
| PPX | NVDM | 19069 | **1731** | 1582 |
| | Bernoulli | **1035** | 1987 | 1161 |
| | NVDM+sig | 21515 | 2266 | **1039** |
| $-\log(p(x|v))$ | NVDM | **442** | **884** | 433 |
| | Bernoulli | **442** | 922 | 446 |
| | NVDM+sig | 445 | 928 | **406** |
| P-NPMI | NVDM | 0.181 | 0.043 | 0.204 |
| | Bernoulli | **0.209** | **0.061** | 0.229 |
| | NVDM+sig | 0.202 | 0.060 | **0.239** |
| NPMI (Avg/S) | NVDM | 0.212/0.226 | 0.138/0.131 | 0.232/0.216 |
| | Bernoulli | 0.217/**0.251** | **0.181**/0.152 | **0.239/0.239** |
| | NVDM+sig | **0.232**/0.248 | 0.174/**0.184** | 0.193/0.219 |

Table 7: 3 Documents provided for 10% of Topics with 1 label per document.

| Metric | Model | Bibtex | Delicious | ArXiv |
|---|---|---|---|---|
| PPX | NVDM | 16839 | **1994** | 1436 |
| | Bernoulli | **946** | 2072 | **1000** |
| | NVDM+sig | 20810 | 2034 | 1044 |
| $-\log(p(x|v))$ | NVDM | 453 | **888** | **428** |
| | Bernoulli | **438** | 923 | 435 |
| | NVDM+sig | 452 | 912 | 401 |
| P-NPMI | NVDM | 0.198 | 0.083 | 0.206 |
| | Bernoulli | 0.206 | 0.073 | 0.235 |
| | NVDM+sig | **0.229** | **0.094** | **0.237** |
| NPMI (Avg/S) | NVDM | **0.240/0.243** | 0.141/0.190 | 0.216/0.206 |
| | Bernoulli | 0.218/0.224 | 0.165/**0.232** | **0.226/0.216** |
| | NVDM+sig | 0.226/0.222 | **0.180**/0.213 | 0.206/0.203 |

Table 8: 10 Documents provided for 10% of Topics with 50% labels per document.

| Metric | Model | Bibtex | Delicious | ArXiv |
|---|---|---|---|---|
| PPX | NVDM | 18254 | **1743** | 1262 |
| | Bernoulli | **965** | 1752 | 1139 |
| | NVDM+sig | 20860 | 2204 | **1052** |
| $-\log(p(x|v))$ | NVDM | 450 | **883** | 428 |
| | Bernoulli | **440** | 902 | 447 |
| | NVDM+sig | 451 | 924 | **402** |
| P-NPMI | NVDM | 0.189 | 0.062 | 0.232 |
| | Bernoulli | 0.191 | 0.049 | 0.210 |
| | NVDM+sig | **0.218** | **0.103** | **0.233** |
| NPMI (Avg/S) | NVDM | **0.239/0.244** | 0.132/0.168 | 0.217/0.222 |
| | Bernoulli | 0.224/0.229 | 0.161/0.153 | **0.219/0.226** |
| | NVDM+sig | 0.230/0.222 | **0.183/0.186** | 0.201/0.211 |

Table 9: 10 Documents provided for 10% of Topics with 100% labels per document.

