# OpenReview forum: "WEAKLY SEMI-SUPERVISED NEURAL TOPIC MODELS"
_ICLR.cc/2019/Workshop/LLD — LLD 2019_

### Official Review · AnonReviewer1 · 2019-04-05
**A reasonable approach to a practical problem**

**Rating:** 3
**Confidence:** 1

**Review:**

Title:
Rating: 3
Confidence: 1

Summary:
The authors present a method of weakly supervising a topic model to guide it toward topic alignments that better align with user intuition for what topics should be. Their approach allows the user to label a subset of documents with a subset of their topics, making it flexible.

I am not in a great position to confirm the soundness of the math, but the problem being solved and the suggested approach seem reasonable and relevant, with practical application.

Points:
- I appreciate the flexibility of the design; allowing any number of documents to be labeled with any number of topics makes it much more likely to be used in practice, I believe.
- The design decision to incorporate this information in the prior is nice and largely decoupled from the VAE in a good way.
- I found the explanation of how ground truth was determined to be a little opaque. If this method in used in other works, please cite them. If it is not, more justification should be given here (possibly with a figure or equation for clarification).
- Figure 1 is very helpful in justifying why you chose the model you did.
- Table 2 does not seem to add much. It is not obvious from looking at the table that any of those methods is better than the others.
- What are the confidence intervals on the results in Table 3? Some of this differences are quite small.

Nits:
- Separate Table 1 from the text more
- You mention in Section 5 that Delicious has 20 topics, but then say that "half" the number of labels is 11, and fully-labeled is up to 28? Please clarify what is being referred to or fix those numbers.

---

### Official Review · AnonReviewer2 · 2019-04-06
**unclear**

**Rating:** 2
**Confidence:** 1

**Review:**

This short paper investigates weak supervision for neural-based topic modelling (NTM) by modifying both the priors and the posteriors, in a VAE setting.

The priors are modified to using the logits of the probability for each topic to appear in a document. The use of the logits is meant to strengthen "ground truth" information given by a user (supervision), while not penalizing the probability for another topics to appear in a document which would not have been labelled by the user (weak supervision).
On the one hand, his part of the work is quite clear, and the authors justify this as a way to make a NMT better align with the semantic of the user.
On the other hand, the explanation about the generative process is unclear, and would benefit of a longer explanation.
I understood that, in addition to the normal setting of a VAE, two other posteriors have been tested: the Concrete approximation to Bernouilli, and the use of a logit-normal distribution.
The all section about the comparison of the two is really confusing and it becomes hard to clearly follow authors' reasoning.

In their experiments, the authors have used three sources of multi-labelled dataset. Topic modelling is not my domain of expertise, but could the authors have not used other datasets to evaluate their approach against (e.g. 20NewsGroups and RCV1-v2 datasets, similarly to original NVDM (Miao et al., 2016))? What's the motivation for using BibTeX, Deliicious and AAPD?
Based on table 3, both the NVDM-o (with logit-normal posterior), and Bernouilli (columns 2 and 3) are quite similar for at least two of the metrics, but this point is not discussed by the authors, at all.

* other comments/questions about the paper *
- latent Dirichlet allocation (LDA) => Latent [...];
- Often times => Oftentimes;
- The integration of table 1 could be better;
- Why is Wikipedia mentioned?
- what's the difference between P-NPMI and NPMI? It's not clear;
- due the => due to the;
- W_{K} => W_{k}  (Bernouilli decoding computing the word distribution), compared to "W_{k} refers to";

---

### Decision · Program_Chairs · 2019-04-16
**Acceptance Decision**

Accept